# Many-Shot In-Context Learning in Multimodal Foundation Models

## Abstract

Large language models are well-known to be effective at few-shot in-context learning (ICL). Recent advancements in multimodal foundation models have enabled unprecedentedly long context windows, presenting an opportunity to explore their capability to perform ICL with many more demonstrating examples. In this work, we evaluate the performance of multimodal foundation models scaling from few-shot to many-shot ICL. We benchmark GPT-4o and Gemini 1.5 Pro across 14 datasets spanning multiple domains (natural imagery, medical imagery, remote sensing, and molecular imagery) and tasks (image classification, visual question answering, and object localization). We observe that many-shot ICL, including up to almost 2,000 multimodal demonstrating examples, leads to substantial improvements compared to few-shot (<100 examples) ICL across all of the datasets. Further, Gemini 1.5 Pro performance continues to improve log-linearly up to the maximum number of tested examples on many datasets. We also find open-weights multimodal foundation models like Llama 3.2-Vision and InternLM-XComposer2.5 do not benefit from the demonstrating examples, highlighting an important gap between open and closed multimodal foundation models. Given the high inference costs associated with the long prompts required for many-shot ICL, we also explore the impact of batching multiple queries in a single API call. We show that batching up to 50 queries can lead to performance improvements under zero-shot and many–shot ICL, with substantial gains in the zero-shot setting on multiple datasets, while drastically reducing per-query cost and latency. Finally, we measure ICL data efficiency of the models, or the rate at which the models learn from more demonstrating examples. We find that while GPT-4o and Gemini 1.5 Pro achieve similar zero-shot performance across the datasets, Gemini 1.5 Pro exhibits higher ICL data efficiency than GPT-4o on most datasets. Our results suggest that many-shot ICL could enable users to efficiently adapt multimodal foundation models to new applications and domains.

## 1 Introduction

Large language models (LLMs) have been shown to substantially benefit from the inclusion of a few demonstrating examples (*shots*) in the LLM context before the test query (Brown et al., 2020; Parnami & Lee, 2022; Wang et al., 2020). This phenomenon, commonly referred to as in-context learning (ICL), enables LLMs to learn from few shots without any updates to model parameters, and therefore improves specialization to new tasks without any further model training. More recently, large multimodal models (LMMs) have also demonstrated the capability of learning from in-context examples Achiam et al. (2023); Han et al. (2023); Zhang et al. (2024b). Han et al. (2023) and Zhang et al. (2024b) both show that few-shot multimodal ICL specifically helps to improve LMM performance on out-domain or out-of-distribution tasks.

While few-shot ICL has enabled promising performance improvements for both LLMs and LMMs, limited model context windows have constrained research on the impact of increasing the number of demonstrating examples on performance. This is especially true for LMMs as most use a large number of visual tokens to represent images. However, due to recent advancements enabling substantially longer context windows – for example, 128,000 tokens for GPT-4o and up to one million tokens for Gemini 1.5 Pro – it is now possible to explore the effect of drastically increasing the number of demonstrating examples.

Figure 1: **Many-shot multimodal in-context learning compared to zero-shot and few-shot multimodal ICL.** In zero-shot and few-shot settings, respectively, no demonstrating examples or only a small number of demonstrating examples are provided in the context before the test query. In a many-shot ICL setting, we include a large number of demonstrating examples in the prompt, whereas in batched many-shot ICL, we perform multiple queries at once using query references.

To investigate the capability of state-of-the-art multimodal foundation models to perform many-shot ICL, we conduct a large suite of experiments benchmarking model performance on 14 datasets spanning several domains and multimodal tasks after scaling up the number of demonstrating examples by multiple orders of magnitude. Specifically, our contributions are as follows:

1. We show that providing close-weights multimodal foundation models with many demonstrating examples leads to substantial performance improvements compared to providing only a few demonstrating examples. We observe that the performance of Gemini 1.5 Pro generally improves log-linearly as the number of demonstrating examples increases, whereas GPT-4o exhibits less stable improvements as the number of in-context examples increases.

2. We find open-weights multimodal foundation models like Llama 3.2-Vision and InternLM-XComposer2.5 do not benefit from the demonstrating examples, highlighting a significant gap and an important direction for the open-weights community.

3. We measure the data efficiency of the models under ICL as the number of demonstrating examples increases, and find that Gemini 1.5 Pro exhibits higher ICL data efficiency than GPT-4o on most datasets.

4. We demonstrate that batching multiple queries into a single request can achieve similar or better performance than single query requests in a many-shot setting, while enabling substantially lower per-example latency and much cheaper per-example inference cost.

5. We find that batching multiple questions can lead to substantial performance improvements in a zero-shot setting. We design experiments to explain this phenomenon, and find that the improvements are due to a combination of domain calibration, class calibration, and self-generated demonstrating examples due to autoregressive decoding.

6. We release a new tool (anonymized link) to allow users to easily test the many-shot ICL capabilities of multimodal foundation models, facilitating future work on studying them.

## 2 RELATED WORK

**Scaling ICL.** The seminal work of Brown et al. (2020) discovered performance improvements for LLMs from increasing the number of in-context examples, but the tested number of demonstrating examples was low (10 to 100), likely due to the restrictive context size (2048 tokens for GPT3).

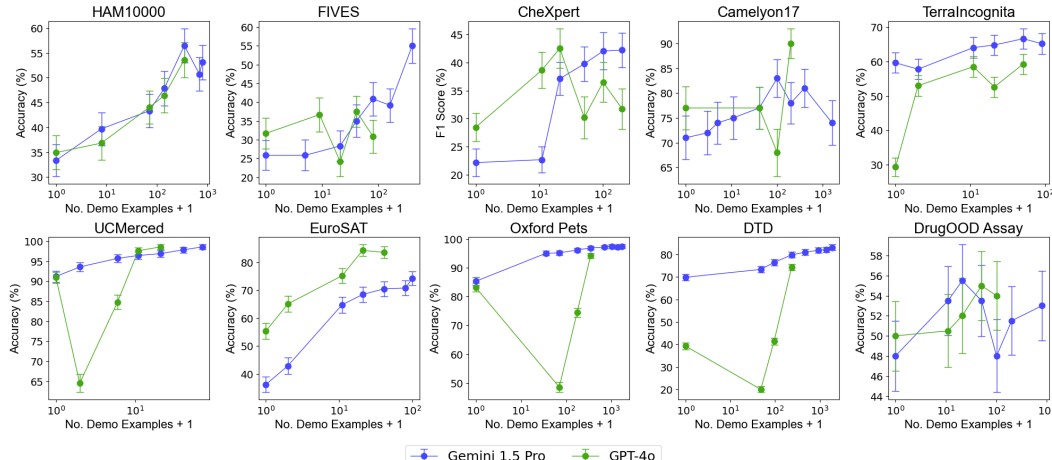

Figure 2: **Gemini 1.5 Pro and GPT-4o performance from zero-shot to many-shot ICL.** The x-axis is in log scale. For Gemini 1.5 Pro, we observe log-linear improvement on 9 out of the 10 datasets and. For GPT-4o, we observe improvement from more demonstrating examples on most datasets, while the improvement is substantially less stable than Gemini 1.5 Pro. Error bars are estimated standard deviations using bootstrapping with 1,000 bootstrap replicates.

Increasing the number of in-context examples has only been explored recently by a few works Li et al. (2023); Agarwal et al. (2024); Bertsch et al. (2024). Both Li et al. (2023) and Agarwal et al. (2024) explore scaling in-context learning to more than 1,000 demonstrating examples and find performance improvements across multiple tasks. However, their experiments are limited to text-only benchmarks and do not compare performance across different models.

**Multimodal ICL.** Due to the recent emergence of LMMs, research on multimodal ICL is still nascent. One prior work developed a new model to leverage complex prompts composed of multimodal inputs in order to allow models to compare images Zhao et al. (2023), while other recent works explored the generalizability of GPT-4V and Gemini to multimodal out-domain and out-of-distribution tasks, and found that ICL leads to performance benefits for both models across many tasks Zhang et al. (2024b); Han et al. (2023). However, none of these works have leveraged the new largely expanded context windows to investigate the effects of increasing the number of demonstrating examples.

**Batch Querying.** Multiple prior works have explored batching queries (also commonly referred to as batch prompting) for more efficient and cheaper inference. Batch prompting was first introduced in Cheng et al. (2023), leading to comparable or better performance than single prompting, while achieving substantially reduced inference token cost and latency. Lin et al. (2023) observe performance degradation with batched prompts in longer contexts, and propose a variety of techniques to mitigate the performance loss. More recently, additional variations of batch prompting have been proposed, including grouping similar questions together Liu et al. (2024), batching prompts of different tasks Son et al. (2024), and concatenating multiple images into a single image collage Xu et al. (2024). We again note that batch prompting with high numbers of demonstrating examples and high numbers of queries has only become feasible due to larger context windows of recent models.

## 3 METHODS

We conduct several experiments to test the effect of increasing the number of demonstrating examples on the performance of state-of-the-art multimodal foundation models: close-weights models like GPT-4o and Gemini 1.5 Pro, and open-weights models like Llama3.2-Vision (Section 3.1). We benchmark their performance using standard performance metrics as well as an ICL data efficiency metric (Section 3.3) on 14 datasets spanning several vision domains and multimodal tasks (Section 3.2). We conduct ablation studies to test the impact of batching queries on model performance and explain the substantial improvement in zero-shot settings (Section 4.2). We refer to the many-

Table 1: **Summary of datasets.** We use 14 datasets spanning multiple domains (natural imagery, medical imagery, remote sensing, and molecular imagery) and tasks (multi-class, multi-label, and fine-grained classification, visual question answering, object localization).

| Dataset | Task and image type | Demo / test set size | Example image |
|---|---|---|---|
| HAM10000 (Tschandl et al., 2018) | 7-category skin disease classification on clinical photos | 805 / 210 |  |
| FIVES (Jin et al., 2022) | 4-category eye disease classification on fundus images | 400 / 120 |  |
| CheXpert (Irvin et al., 2019) | Multi-label 5-category lung disease detection on chest X-rays | 200 / 150 |  |
| Camelyon17 (Bandi et al., 2018) | Binary tumor detection on pathology images | 2000 / 100 |  |
| TerraIncognita (Beery et al., 2018) | 9-category animal species recognition on camera images | 1035 / 270 |  |
| UCMerced (Yang & Newsam, 2010) | 21-category land use classification on satellite images | 1470 / 420 |  |
| EuroSAT (Helber et al., 2019) | 10-category land use / land cover classification on satellite images | 1000 / 300 |  |
| Oxford Pets (Parkhi et al., 2012) | 35-category pet classification on camera images | 1750 / 700 |  |
| DTD (Cimpoi et al., 2014) | 47-category texture classification on synthetic images | 2350 / 940 |  |
| DrugOOD Assay (Ji et al., 2022) | Binary drug binding prediction on molecular images | 1600 / 200 |  |
| RSVQA (Lobry et al., 2020) | Visual question answering on satellite images | 200/200 |  |
| VQA-RAD (Lau et al., 2018) | Visual question answering on radiology images | 200/200 |  |
| DIOR (Li et al., 2020) | Object localization on satellite images | 200/100 |  |
| DeepLesion (Yan et al., 2018) | Lesion localization on CT images | 200/100 |  |

shot in-context learning framework as many-shot ICL. Figure 1 provides an illustrative summary of many-shot ICL and batched many-shot ICL compared to zero-shot and few-shot ICL.

## 3.1 MODELS

We use three state-of-the-art multimodal foundation models with public API access, namely GPT-4o, GPT4(V)-Turbo (Achiam et al., 2023), and Gemini 1.5 Pro (Reid et al., 2024). In addition, two open-weights multimodal foundation models (Llama3.2-11B-Vision (lla) and InternLM-XComposer-2.5 (Zhang et al., 2024a)) are also tested. Because GPT-4o performs substantially better than GPT4(V)-Turbo, we focus on the results of GPT-4o and Gemini 1.5 Pro in the main text, and include GPT4(V)-Turbo results in the Appendix. We do not utilize Claude3-Opus in our experiments, as it only accepts up to 20 images in one request at the time of writing. The specific endpoint for for GPT-4o is "gpt-4o-2024-05-13", for GPT-4(V)-Turbo is "gpt-4-turbo-2024-04-09", and for Gemini 1.5 Pro is "gemini-1.5-pro-preview-0409". We use the API service provided by OpenAI for GPT-4o and GPT-4(V)-Turbo, and the API service provided by Google Cloud on Vertex AI for Gemini 1.5 Pro. We set the temperature to zero for all models and a random seed for GPT-4(V)-Turbo and GPT-4o to obtain more deterministic responses. To prevent models from abstaining (which happens rarely), we rerun the query until an answer is provided. We call the APIs on a virtual machine instance of type "c2-standard-8" and run inference using open-weights models on a "a3-highgpu-8g" machine (with 8 H100 GPUs) hosted on Google Cloud Platform.

## 3.2 DATASETS

We benchmark the model performance on 14 datasets spanning multiple domains (natural imagery, medical imagery, remote sensing, and molecular imagery) and tasks (image classification, visual question answering, and object localization). We acknowledge most LMMs are not yet capable of accurately producing localizations (Wu et al., 2024; Zang et al., 2023). Table 1 provides a summary of the datasets used in this study.

For all datasets, we construct a set of demonstrating (demo) examples from the original training and validation splits used for in-context learning and a test set from the original test split (if one exists) to evaluate the performance of the models. We randomly sample the demo and test sets from the original dataset without replacement. For the multi-class and fine-grained classification datasets, we perform a class-stratified sampling, ensuring an equal number of examples per class in both the demo and test sets. For the multi-label classification dataset (CheXpert), we sample an equal number of positive and negative samples per class in both the demo and test sets. We note that, since the task is multi-label, this sampling procedure does not result in an exactly equal number of examples per class. For the scaling experiments, we increase the number of demonstrating examples while ensuring class balance. Besides the 10 classification datasets shown in Table 1, we also include two visual question answering datasets (RSVQA (Lobry et al., 2020) and VQA-RAD (Lau et al., 2018)) and two object localization datasets (DIOR for bridge localization (Li et al., 2020) and DeepLesion for lung lesion localization (Yan et al., 2018)).

## 3.3 EVALUATION METRICS

We use standard metrics to evaluate model performance on each dataset. Specifically, we measure performance using accuracy for all multi-class classification and visual question answering datasets as they are sampled to have a balanced class distribution. For multi-label classification on CheXpert, we use the macro-averaged F1 metric. In the rare case of parsing errors, we consider the response as incorrect. For object localization datasets, we use mean Intersection over Union(IoU). To estimate the variability around the evaluation metrics, we compute standard deviation using bootstrapping with 1,000 bootstrap replicates, and it captures the variability in data sampling.

In addition to standard performance metrics, we measure the data efficiency of each model. Specifically, we compute a linear regression between $\log_{10}(N + 1)$ (with $N$ the number of examples) and model performance, enforcing that the line passes through the zero-shot performance point. This value approximates the amount of performance improvement from zero-shot expected from including an order of magnitude more demonstrating examples.

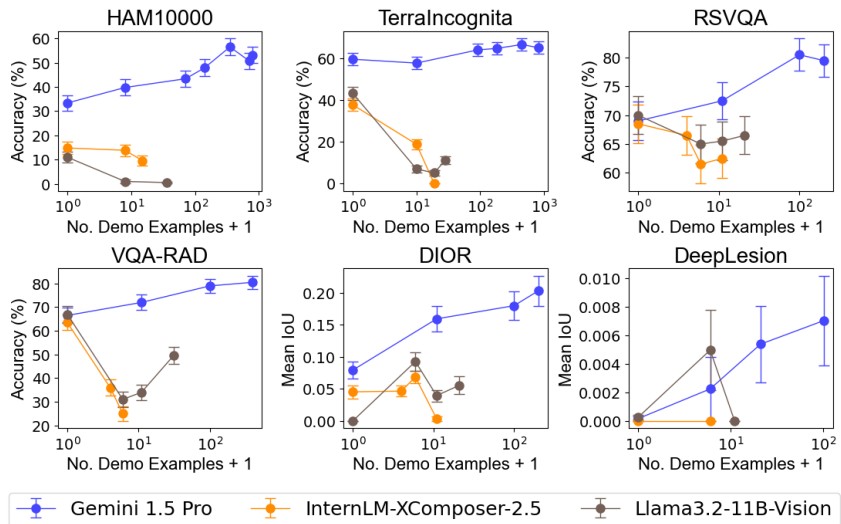

Figure 3: **Llama3.2 and InternLM-XComposer2.5 performance compared with Gemini 1.5 Pro.** While Gemini 1.5 Pro shows substantial improvement from zero-shot to many-shot ICL on all three categories of tasks, both open-weights models do not benefit from demonstrating examples.

# 4 RESULTS

We present many-shot ICL performance using batched queries in Section 4.1, investigate the impact of batching queries on performance in Section 4.2, and provide an analysis on cost and latency in Section 4.3. Results using GPT4(V)-Turbo are in Appendix C.

## 4.1 INCREASING NUMBER OF DEMONSTRATING EXAMPLES

**Main Results.** Gemini 1.5 Pro exhibits consistent and substantial improvements as the number of demonstrating examples increases across all datasets except for DrugOOD Assay (Figure 2 and 3). Gemini 1.5 Pro shows particularly large improvements from many-shot ICL on HAM10000 (+23% accuracy compared to zero-shot, +16% compared to 7 examples), FIVES (+29% compared to zero-shot, +27% compared to 20 examples), and EuroSAT (+38% compared to zero-shot, +31% compared to 10 examples). Notably, for 8 out of the 14 datasets (FIVES, UCMerced, EuroSAT, Oxford Pets, DTD, VQA-RAD, DIOR and DeepLesion), Gemini 1.5 Pro performance continues to improve up to the highest number of demonstrating examples considered (~1,000 examples). On the other 6 datasets, the optimal performance occurs prior to the highest number of demo examples, with the maximum number of demo examples leading to similar or slightly worse performance than the optimal demo set size. On the other hand, Gemini 1.5 Pro performance on DrugOOD Assay does not substantially benefit from many-shot ICL, with high variance in performance across demo sizes and the peak performance at 40 demo examples.

Similarly, GPT-4o shows substantial performance improvements on all datasets except FIVES and DrugOOD Assay using many-shot ICL, but the improvement is not consistent. For many datasets, performance drops sharply at first and then improves significantly as the number of demonstrating examples increases further, resulting in V-shaped scaling curves (Figure 2). We also note that we were unable to increase the number of demo examples to the same level as considered for Gemini 1.5 Pro because GPT-4o has a shorter context window and is more prone to timeout errors with longer inputs. GPT-4o performance on DrugOOD Assay shows high variance, similar to Gemini 1.5 Pro, with the peak performance observed at 50 demo examples.

**Open-weights Model Results.** We also find open-weights multimodal foundation models like Llama 3.2-Vision and InternLM-XComposer2.5 do not benefit from the demonstration examples (Figure 3), highlighting a significant gap between open and closed multimodal foundation models.

Table 2: **Many-shot ICL performance and efficiency comparison.** We report the performance under a zero-shot regime and performance at the optimal demo set size as well as the many-shot ICL data efficiency of GPT-4o and Gemini 1.5 Pro. We measure performance using accuracy on all datasets except CheXpert, for which we use macro-averaged F1. The highest ICL data efficiency between the two models on each dataset is marked in bold.

| Dataset | GPT-4o | | | Gemini 1.5 Pro | | |
|---|---|---|---|---|---|---|
| | Zero-shot | Best | Efficiency | Zero-shot | Best | Efficiency |
| HAM10000 | 34.93 | 53.59 (+18.66) | 5.91 | 33.33 | 56.46 (+23.13) | **6.94** |
| FIVES | 31.67 | 37.50 (+5.83) | 0.30 | 25.83 | 55.00 (+29.17) | **7.56** |
| CheXpert | 28.47 | 42.54 (+14.08) | 3.70 | 22.16 | 42.23 (+20.08) | **9.06** |
| Camelyon17 | 77.00 | 90.00 (+13.00) | 1.00 | 71.00 | 83.00 (+12.00) | **3.00** |
| TerraIncognita | 29.26 | 59.26 (+30.00) | **20.50** | 59.63 | 66.67 (+7.04) | 3.50 |
| UCMerced | 90.95 | 98.57 (+7.62) | 1.20 | 91.19 | 98.57 (+7.38) | **4.36** |
| EuroSAT | 55.37 | 84.23 (+28.86) | 19.40 | 36.24 | 74.16 (+37.92) | **20.61** |
| Oxford Pets | 83.14 | 94.14 (+11.00) | -3.72 | 85.29 | 97.43 (+12.14) | **4.26** |
| DTD | 39.26 | 74.47 (+35.21) | **4.48** | 69.89 | 83.19 (+13.30) | 3.89 |
| DrugOOD Assay | 50.00 | 55.00 (+5.00) | 2.02 | 48.00 | 55.50 (+7.50) | **2.03** |

**Sensitivity to prompt selection.** We also explore a different set of prompts to test the robustness of many-shot ICL to differences in prompt wording on two datasets. While there is a small deviation in performance between different prompts, the overall log-linear improvement trend is consistent across the prompts. Details can be found in Appendix B.

**ICL data efficiency.** We find Gemini 1.5 Pro demonstrates higher ICL data efficiency than GPT-4o across all datasets except TerraIncognita and DTD (Table 2). Gemini 1.5 Pro ICL efficiency is especially high on EuroSAT, with 20.61% improvement in accuracy for every 10x more demo examples, and lowest on DrugOOD Assay (2.03), Camelyon17 (3.00), and TerraIncognita (3.50). GPT-4o ICL data efficiency is especially high on TerraIncognita (20.50%) and EuroSat (19.40). Gemini 1.5 Pro has a positive efficiency on all datasets and GPT-4o has a positive data efficiency on 9 of the 10 datasets (excluding Oxford Pets). Importantly, both models benefit substantially from many-shot ICL at the optimal demo set size, with an average improvement of +17% for both Gemini 1.5 Pro and GPT-4o.

### 4.2 IMPACT OF BATCHING QUERIES

As including a large set of demo examples in the prompt leads to much longer sequence lengths and therefore higher inference time and cost, we consider batching queries in a single prompt to reduce per-query cost, and examine the impact of different batch sizes on model performance. Due to its superior performance and free preview access, we use Gemini 1.5 Pro for these experiments.

**Main Results.** We find minimal performance degradations, and sometimes performance improvements, as we increase the number of queries included in each batch across under both zero-shot and many-shot (at the optimal demo set size) regimes (Figure 4). Notably, using a single query each time with many-shot ICL is suboptimal across many of the datasets. We find that the optimal batch size is among the three largest sizes on every dataset except CheXpert and EuroSAT, which both see optimal performance with a single query at a time.

We additionally observe that including a single query at a time is suboptimal on most datasets in the zero-shot regime. Surprisingly, performance with the highest batch size is substantially higher across three datasets under the zero-shot regime, with a consistent performance improvement as the batch size is increased on both UCMerced and Terraincognita.

**Zero-shot performance improvements from batching queries.** We conduct several additional experiments to investigate why batch querying can lead to large performance improvements under the zero-shot regime on TerraIncognita and UCMerced. We hypothesize that this improvement may be due to three potential benefits from ICL: (1) domain calibration, where the model benefits from seeing more images in the domain in order to adapt to it, (2) class calibration, where seeing

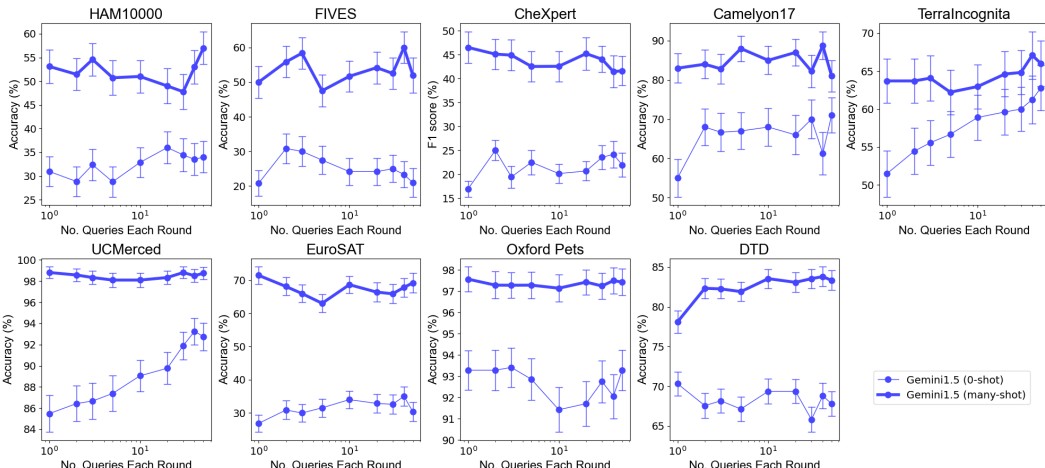

Figure 4: **Gemini 1.5 Pro performance under many-shot and zero-shot ICL with varying amount of queries included in every request.** We show performance per batch size with the optimal number of demo examples (many-shot) and no demo examples (zero-shot). The $x$-axis is in log scale. Under the many-shot regime, batching queries leads to no substantial drop in performance compared to individual queries when we choose a suitable batch size. For zero-shot, including only one query is suboptimal for many datasets. Error bars are estimated standard deviations using bootstrapping with 1,000 bootstrap replicates.

images of different classes enables the model to better calibrate its outputs(Min et al., 2022), and (3) self-ICL (shown to be effective in prior work Chen et al. (2023)), where the model can learn from self-generated demonstrations due to autoregressive decoding. We design experiments to isolate the potential benefits from each of these types of ICL between asking a single query to batching 50 queries together.

First, to measure potential improvement from domain calibration, we include 49 images from the same class in the prompt without including any label. We find a 3.0% improvement on TerraIncognita and 2.6% degradation on UCMerced, suggesting domain calibration is helpful for the former but not the latter. Second, to capture performance gains from class calibration, we include a random sample of 49 images in the prompt, again without including the label. We see a further 3.5% improvement on TerraIncognita (6.5% improvement from a single query) and a 4.5% improvement from a single query on UCMerced, suggesting including the context of class-balanced images is helpful even without labels. Third, to capture additional performance improvements from the self-generated labels, we obtain predicted labels from the zero-shot model using a single query for each of the 49 randomly sampled images and add them to the prompt. We observe further performance increase on both datasets, with 5.5% on TerraIncognita and 2.7% on UCMerced. The final total accuracy is similar to asking the 50 questions each round, which suggests these three components mostly explain the reason for improved zero-shot performance under a larger query batch size.

### 4.3 COST AND LATENCY ANALYSIS

Many-shot ICL incurs zero additional training cost, but per-query inference can be costly and slow due to long input contexts. To quantitatively measure this, we compute the latency and cost associated with the zero-shot and many-shot requests with and without batching when using Gemini 1.5 Pro on HAM10000 and TerraIncognita. We calculate the costs using the Gemini 1.5 Pro preview pricing ($7 per 1 million input tokens and $21 per 1 million output tokens). For fair comparison and to minimize data transfer artifacts, all requests are sent to the same location where the VM instance is held ("us-central1"). We run the query three times under each setting and report the average.

In the zero-shot regime, we see substantial per-example latency reductions due to query batching, close to a 10x reduction on HAM10000 and 2x on TerraIncognita (Table 3). The per-example cost is similar between the two as there is no additional context needed for including demonstrating examples. In the many-shot regime, we observe substantial reductions in both per-example latency

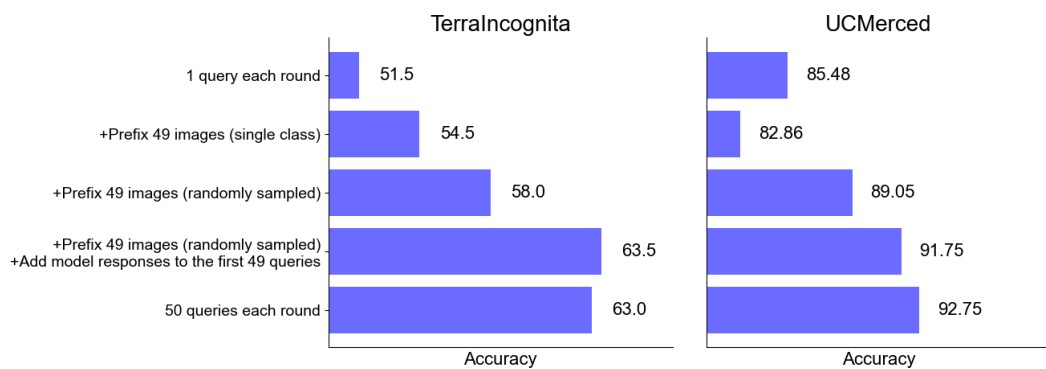

Figure 5: **Ablation study to investigate why batching queries leads to performance improvements when using Gemini 1.5 Pro in a zero-shot setting.** The first bar shows performance when including a single query, the second adds 49 unlabeled images from a single class, the third adds 49 unlabeled images in total from all classes, the fourth adds model responses to include self-generated demonstrations, and the last includes 50 queries in one request.

and cost on both datasets. Specifically, for HAM10000, we find a near 35x reduction in latency and 10x reduction in cost, and 20x reduction in latency and 45x reduction in cost for TerraIncognita.

## 5 DISCUSSION

In this study, we evaluate many-shot ICL of state-of-the-art multimodal foundation models across 14 datasets and find consistent performance improvements across most of the datasets. Batching queries with many-shot ICL further exhibits substantially reduced per-example latency and inference costs without compromising performance. We also highlight one future direction for improving open-weights multimodal foundation models as they are unable to learn from demonstrating examples.

Our findings suggest that these multimodal foundation models have the capability of performing ICL with large numbers of demonstrating examples, which may have significant implications on their practical use. For example, it was previously impossible to adapt these large, private models to new tasks and domains, but many-shot ICL would enable users to leverage demonstrating examples to adapt the models. One significant advantage of many-shot ICL is its ability to get quick results even on the same day of model release, and that's why we can finish our evaluation using GPT-4o and Llama3.2 within days. Furthermore, fine-tuning open-source models is the standard practice when practitioners have access to moderately sized datasets, but many-shot ICL may remove the need for fine-tuning, making it much easier to develop customized approaches. We note that it remains to be seen how traditional fine-tuning of these models compares to many-shot ICL with foundation models in terms of absolute performance and data efficiency, so future work should explore this. In addition, it is important to study general issues which plague those foundation models, such as hallucinations and biases, under the context of many-shot ICL and batching queries. For example, it would be interesting to explore if carefully curated and large sets of demonstrating examples can reduce biases across different sub-groups. We leave this to future work.

Our study has limitations. First, we only explore performance under many-shot ICL on some common multimodal tasks which we believe are the most practically relevant and common multimodal settings, but it is worthwhile for future work to explore potential benefits from many-shot ICL on other tasks such as VL-ICL Bench Zong et al. (2024). Second, even after recent developments to increase context size, the size prohibits many-shot ICL from being used on datasets with a large number (several hundred or more) of classes. We anticipate that context window sizes will continue to increase in size over time which will mitigate this issue. Third, we only run the ablation experiments on Gemini 1.5 Pro due to budget limit, but it will be interesting to whether these trends hold for GPT-4o. Fourth, the datasets which were used to train these private models have not been disclosed, so it is difficult to tell whether the models have been trained on the datasets we selected. We argue that the zero-shot performance across the datasets is far from perfect, which suggests that the datasets have not been used for training, but we cannot determine that with certainty.

Table 3: **Inference latency and cost using Gemini 1.5 Pro with and without query batching.** We use 50 queries per batch. In the zero-shot setting, we can achieve lower per-example latency with batching, but the per-example cost remains identical. In the many-shot setting, the per-example cost and per-example latency both drop substantially with query batching.

| Dataset | Without Query Batching | | | With Query Batching | | |
|---|---|---|---|---|---|---|
| | Per-batch Latency | Per-example Latency | Per-example Cost | Per-batch Latency | Per-example Latency | Per-example Cost |
| HAM10000 (zero-shot) | 2.2s | 2.2s | $0.0038 | 11.4s | 0.23s | $0.0038 |
| TerraIncognita (zero-shot) | 2.0s | 2.0s | $0.0037 | 51.6s | 1.0s | $0.0038 |
| HAM10000 (350-shot) | 17.3s | 17.3s | $0.8420 | 26.9s | 0.54s | $0.0877 |
| TerraIncognita (810-shot) | 34.9s | 34.9s | $1.8420 | 85.9s | 1.7s | $0.0406 |

## 6 CONCLUSION

In summary, we show that state-of-the-art multimodal foundation models, especially Gemini 1.5 Pro, are capable of many-shot ICLto achieve substantial performance improvements with cost efficiency across multiple domains and tasks. However, open-weights models like Llama 3.2-Vision do not exhibit the same benefits, highlighting a significant future direction. We believe that these results pave a promising path forward to improve the adaptability and accessibility of large multimodal foundation models.

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

# A PROMPTS USED FOR ICL EXPERIMENTS

## A.1 PROMPT USED FOR IMAGE CLASSIFICATION EXPERIMENTS

```
prompt = ""
for demo in demo_examples:
    prompt += f"""<>Given the image above, answer the following question-
using the specified format.
Question: What is in the image above?
Choices: {str(class_desp)}
Answer Choice: {demo.answer}
"""

prompt += f"""<>Given the image above, answer the following question-
using the specified format.
Question: What is in the image above?
Choices: {str(class_desp)}

Please respond with the following format:
---BEGIN FORMAT TEMPLATE---
Answer Choice: [Your Answer Choice Here]
Confidence Score: [Your Numerical Prediction Confidence Score Here From 0 To 1]
---END FORMAT TEMPLATE---

Do not deviate from the above format. Repeat the format template for the answer."""
```

## A.2 PROMPTS USED FOR IMAGE CLASSIFICATION EXPERIMENTS WITH BATCHING

```
prompt = ""
for demo in demo_examples:
    prompt += f"""<>Given the image above, answer the following question-
using the specified format.
Question: What is in the image above?
Choices: {str(class_desp)}
Answer Choice: {demo[1]}
"""

for idx, i in enumerate(test_df.iloc[start_idx:end_idx].itertuples()):
    prompt += f"""<>Given the image above, answer the following question-
using the specified format.
Question {qn_idx}: What is in the image above?
Choices {qn_idx}: {str(class_desp)}

"""

for i in range(start_idx, end_idx):
    qn_idx = i-start_idx+1
    prompt += f"""
Please respond with the following format for each question:
---BEGIN FORMAT TEMPLATE FOR QUESTION {qn_idx}---
Answer Choice {qn_idx}: [Your Answer Choice Here for Question {qn_idx}]
Confidence Score {qn_idx}: [Your Numerical Prediction Confidence Score Here-
From 0 To 1 for Question {qn_idx}]
---END FORMAT TEMPLATE FOR QUESTION {qn_idx}---

Do not deviate from the above format. Repeat the format template for the answer."""
```

### A.3 PROMPTS USED FOR BATCHING ABLATION EXPERIMENTS

#### A.3.1 PREFIXING IMAGES

```
prompt = ""
for demo in prefix_image_paths:
    prompt += f"""<>

"""
prompt += "Above are some images from the same dataset. "
qns_idx = []
for idx, i in enumerate(test_df.iloc[start_idx:end_idx].itertuples()):
    qn_idx = idx+1
    prompt += f"""<> Given the image above, answer the following question-
using the specified format.
Question {qn_idx}: What is in the image above?
Choices {qn_idx}: {str(class_desp)}

"""
for i in range(start_idx, end_idx):
    qn_idx = i-start_idx+1
    prompt += f"""
Please respond with the following format for each question:
---BEGIN FORMAT TEMPLATE FOR QUESTION {qn_idx}---
Answer Choice {qn_idx}: [Your Answer Choice Here for Question {qn_idx}]
Confidence Score {qn_idx}: [Your Numerical Prediction Confidence Score Here-
From 0 To 1 for Question {qn_idx}]
---END FORMAT TEMPLATE FOR QUESTION {qn_idx}---

Do not deviate from the above format. Repeat the format template for the answer."""
```

### A.4 PROMPT USED FOR VISUAL QUESTION ANSWERING EXPERIMENTS

```
prompt = "You're an expert in answering questions on " + ("radiology"/"satellite")-
+ " images. " + ("Here are some demonstration examples: "-
if num_shot_per_class>0 else "")
for demo in demo_examples:
    prompt += f"""<>Given the image above, answer the following question using-
the specified format.
Question: {demo[1]}
Answer: {demo[2]}

"""

for idx, i in enumerate(test_df.iloc[start_idx:end_idx].itertuples()):
    qn_idx = idx + 1

    prompt += f"""<>Given the image above, answer the following question using-
the specified format.
Question {qn_idx}: {i.question}
Choices {qn_idx}: {str(i.choices)}
"""
for i in range(start_idx, end_idx):
    qn_idx = i - start_idx + 1
    prompt += f"""
Please respond with the following format for each question:
---BEGIN FORMAT TEMPLATE FOR QUESTION {qn_idx}---
Answer {qn_idx}: [Your Answer Here for Question {qn_idx}]
---END FORMAT TEMPLATE FOR QUESTION {qn_idx}---
```

```
Do not deviate from the above format. Repeat the format template for the answer."""
```

### A.5 PROMPT USED FOR OBJECT LOCALIZATION EXPERIMENTS

```
prompt = ("Here are some demonstration examples:\n" if num_shot_per_class>0 else "")
for demo in demo_examples:
    prompt += f"""<> Return bounding box for the {object_name} in the above
image with this format: [ymin, xmin, ymax, xmax]
{demo[1]}\n
"""

for idx, i in enumerate(test_df.iloc[start_idx:end_idx].itertuples()):
    qn_idx = idx + 1

    prompt += f"""<> Return bounding box for the {object_name} in the above
image with this format: [ymin, xmin, ymax, xmax]"""
```

## B PROMPT SELECTION

We utilize a different set of prompts to test the robustness of ManyICL to differences in prompt wording. We randomly sample two datasets (HAM10000 and EuroSAT) for this experiment due to budget limit.

### B.1 PROMPTS USED FOR PROMPT SELECTION EXPERIMENTS

Note that only the question section is shown here, and prompt 1 is used for all other image classification experiments.

#### B.1.1 PROMPT 1

```
<>Given the image above, answer the following question using the specified forma
Question {qn_idx}: What is in the image above?
Choices {qn_idx}: {str(class_desp)}
```

#### B.1.2 PROMPT 2

```
<>Given the image above, answer the following question using the specified forma
Question {qn_idx}: Which class does this image belong to?
Choices {qn_idx}: {str(class_desp)}
```

#### B.1.3 PROMPT 3

```
Question {qn_idx}: <>Classify the image above, choose from {str(class_desp)}
```

### B.2 PROMPT SELECTION RESULTS

Figure 6 shows the sensitivity of performance to prompt selection on two datasets with three prompts. While there exists a small deviation in performance, but the overall log-linear improvement trend is consistent.

## C GPT4(V)-TURBO PERFORMANCE UNDER MANY-SHOT ICL

GPT4(V)-Turbo shows mixed results for many-shot ICL, with substantial performance improvements on HAM1000, UCMerced, EuroSAT, and DTD, but minimal improvements or no improvement across the other six datasets (Figure 7). However, we note that we were unable to increase the number of demo examples to the same level as Gemini 1.5 Pro because GPT4(V)-Turbo has a

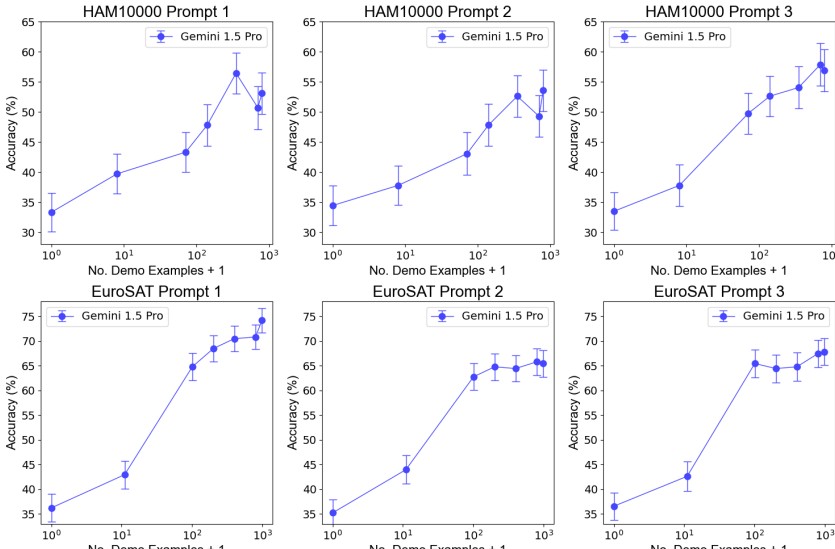

Figure 6: **Sensitivity analysis of many-shot ICL.** These plots show the change in task performance on two datasets as the number of demonstrating examples increases, using three different prompts. For all experiments on sensitivity analysis, the Gemini 1.5 Pro model is used. The $x$-axis is in the logarithmic scale, representing the number of demonstrating examples plus one. The log-linear improvement until the optimal performance is consistent across all prompts selected.

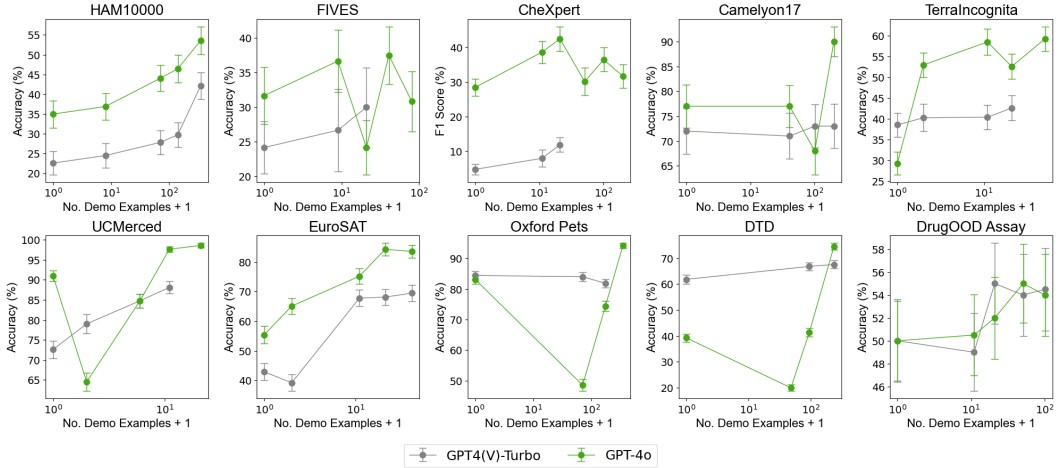

Figure 7: **GPT4(V)-Turbo and GPT-4o performance from zero-shot to many-shot ICL.** We show the accuracy of GPT4(V)-Turbo and GPT-4o as we vary the number of demonstrating examples across the 10 classification datasets. The $x$-axis is in log scale. HAM10000 and TerraIncognita exhibit a relatively smooth log-linear improvement for both models. On the UCMerced, Oxford Pets, and DTD datasets, GPT-4o displays a large drop in performance before reaching the best performance with the largest number of demonstrating examples.

shorter context window and is more prone to timeout errors when scaling. Additionally, GPT4(V)-Turbo seems to generally underperform Gemini 1.5 Pro across the datasets excluding FIVES and EuroSAT for which it seems to mostly match the Gemini 1.5 Pro performance. GPT4(V)-Turbo performance on DrugOOD Assay shows high variance, resembling that of Gemini 1.5 Pro with the peak performance at 40 demo examples.

# D PERFORMANCE OF MANY-SHOT ICL ON MEDICAL QA TASKS

## D.1 PROMPT USED FOR MEDICAL QA EXPERIMENTS (MEDQA, MEDMCQA)

```
prompt = "You are an expert in answering medical exam questions. "
for demo in demo_examples:
    prompt += f"""Question: {demo.question}
Choices: {demo.options}
Answer: {demo.answer}
"""

prompt += f"""Question: {actual.question}
Choices: {actual.options}

Please respond with the following format:
---BEGIN FORMAT TEMPLATE---
Answer: [Your Answer Choice Here]
Confidence Score: [Your Numerical Prediction Confidence Score Here From 0 To 1]
---END FORMAT TEMPLATE---

Do not deviate from the above format. Repeat the format template for the answer."""
```

## D.2 RESULTS

Figure 8 shows the results on medical QA tasks.

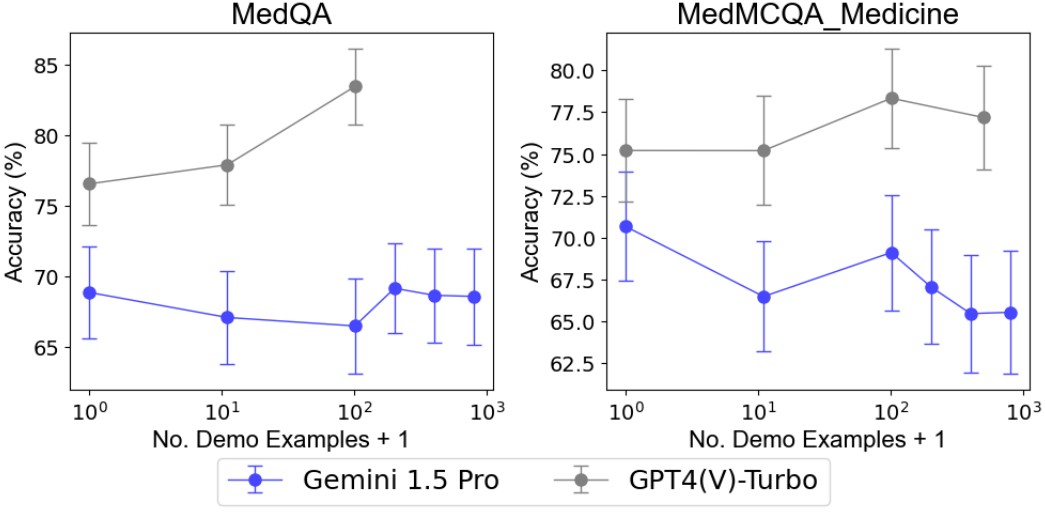

Figure 8: **Many-shot ICL performance of medical QA tasks.** The $x$-axis is in log scale. GPT4(V)-Turbo consistently shows better performance compared to Gemini 1.5 Pro. The accuracy tends to increase for GPT4(V)-Turbo, but Gemini 1.5 Pro performance is more variable.

