# OpenReview forum: "Many-Shot In-Context Learning in Multimodal Foundation Models"
_ICLR.cc/2025/Conference — Submitted to ICLR 2025_

### Official Review · Reviewer_RFrU · 2024-11-01

**Soundness:** 3
**Presentation:** 2
**Contribution:** 3
**Rating:** 5
**Confidence:** 3

**Summary:**

The study assesses the performance of multimodal foundation models in context learning from few-shot to many-shot, comparing the performance of GPT-4o and Gemini 1.5 Pro across 14 datasets in different domains and tasks. It is found that many-shot ICL led to significant improvements, with Gemini 1.5 Pro demonstrating higher ICL data efficiency on most datasets. Additionally, multimodal foundation models with open weights did not benefit from demonstration examples, highlighting the gap between open and closed models. The research also explored the impact of batch processing multiple queries, showing performance improvements and cost reductions under zero-shot and many-shot ICL.

**Strengths:**

1.The authors reveal the performance trends of LMMs as the number of instances increases, noting that many-shot ICL can significantly enhance open-weighted multimodal models.
2. The work conduct extensive experiments on 14 datasets spanning different domains and multimodal tasks to evaluate model performance from various perspectives. This includes comparing the inference speed after employing batch query techniques.
3. In order to mitigate the increase in inference speed with the number of instances, this study utilize batch query techniques to reduce the inference speed and unveiled the impact of number of queries on network model performance with batch query technology.

**Weaknesses:**

1. The introduction of this paper is simple. The benefits of many-shot ICL for LLMs have been well-documented in some works, and this work merely briefly highlights the differences from these works in the related works section, failing to adequately showcase the unique advantages of this work.
2. Although the work reveals that open-weighted models can achieve better gains than closed LMMs as the number of instances changes, it does not explain the reasons behind this phenomenon.
3. The work primarily focuses on model performance but lacks an in-depth analysis. The authors do not provide explanations for why closed LLMs perform poorly on certain datasets while others perform well. They merely present the corresponding experimental results.

**Questions:**

For open-weighted multimodal models that do not benefit from many-shot ICL, can improvements be proposed to enhance their performance in many-shot ICL?
The work mainly focuses on GPT-4o and Gemini 1.5 Pro closed LMMs. How to choose the appropriate model for Many-Shot ICL?

---

### Official Review · Reviewer_JF9j · 2024-11-04

**Soundness:** 2
**Presentation:** 2
**Contribution:** 2
**Rating:** 3
**Confidence:** 3

**Summary:**

This work examines the capabilities of multimodal foundation models in the context of many-shot in-context learning (ICL). The authors evaluated both proprietary and open-source models using 14 datasets spanning multiple domains. Given the high inference costs associated with many-shot ICL, the study also explores the potential of batching multiple queries within a single API call. The findings reveal that batch calling not only reduces latency but also enhances model performance.

**Strengths:**

1. The authors conducted tests on domains that are not commonly used for benchmarking multimodal LLMs, such as medical and remote sensing. While prior studies have explored these areas, there has been relatively less compared to the natural image domain.

2. The authors also address inference efficiency, a major concern when introducing many-shot in-context learning (ICL).

3. Error bars are presented in most of the figures.

**Weaknesses:**

1. Lack of In-Depth Analysis: The authors point out that open-source mLLMs struggle to scale effectively with an increasing number of in-context examples, whereas proprietary models handle this scaling better. However, this discrepancy could simply be attributed to differences in model size. This explanation is not fully explored in the paper, and a deeper analysis would make the argument more compelling.

2. Concerns About Cross-Domain Adaptation: The paper uses many-shot ICL to evaluate LLM performance on cross-domain adaptation (stated in the opening). However, it is unclear whether proprietary models have already been pretrained on these datasets (as acknowledged in the limitations section). This raises the possibility of data leakage, which could explain the performance gap between open-source and proprietary models. Furthermore, it remains uncertain whether the paper's main claim—that cross-domain adaptation can be mitigated by many-shot ICL—holds true, as some supposedly out-of-domain datasets may actually be in-domain for these proprietary models. I suggest the authors only test on open source models that are more transparent with their visual instruction tuning datasets.

3. Benchmark Datasets: While the benchmark includes two small Visual Question Answering (VQA), it primarily focuses on image classification tasks/objection detection that are relatively simple for classic computer vision models, many of which already achieve 95% or 100% accuracy (eurosat, UCmerced). Improving the diversity of the tasks will improve the solidity of the claim.

4. Minor issue: Oxford pets is the dataset from conventional CV domain.

**Questions:**

1. Figure 5: The figure is not referenced in the text.

2. Around Line 368: Do the authors have any hypothesis as to why CheXpert and EuroSAT do not scale up with the number of queries? Providing more insight or potential explanations would strengthen the analysis.

3. Hyper-Parameters for Inference: It would be beneficial for the authors to include details on the hyper-parameters used during inference, such as temperature settings, to enhance the reproducibility of the experiments.

4. Image Classification in A.1: How are the negative choices constructed for the image classification task? Are they sampled from other class names in an offline manner or generated dynamically in an online manner?

---

### Official Review · Reviewer_nSBc · 2024-11-04

**Soundness:** 2
**Presentation:** 3
**Contribution:** 2
**Rating:** 5
**Confidence:** 3

**Summary:**

The authors assess the many-shot in-context learning (ICL) capabilities of state-of-the-art multimodal foundation models across 14 datasets, discovering consistent performance enhancements in the majority of them. Additionally, employing batching with many-shot ICL significantly lowers per-example latency and inference costs, all while maintaining performance levels.

**Strengths:**

1.The paper provides a comprehensive evaluation of many-shot ICL across multiple datasets, showcasing the model's performance in various domains and tasks, with a rigorous research methodology.

2.The authors suggest that batching multiple queries can result in significant performance improvements in zero-shot settings. They design experiments to elucidate this phenomenon from various perspectives, including domain calibration and class calibration. These findings could have important implications for practical applications.

**Weaknesses:**

1.Balanced sampling seems very important, but what if we test with random sampling or other class-wise sampling?

2.The paper explores query batching using examples from the same dataset. What if we batch images from different datasets together to explore the impact of cross-dataset queries?

3.We can observe the scaling performance with the increasing amount of the ICL. In addition to quantity,  does the quality matter? It is great to see the presented results on close-weight and open-weights LLM. Yet, Since the community did research on the ICL for a long while, I was also expecting a more theoretical or more convincing discussion and analysis on your findings.

4.Presentation is straightforward and easy to understand, I also think it is good to see such explorations on LMM. But, on the other hand, I did concern whether such study is qualified for a ICLR paper or just being a technical report? BTW, I cannot see any reference to Figure 5 in the paper.

**Questions:**

Please refer to weakness.

---

### Official Review · Reviewer_DvdU · 2024-11-04

**Soundness:** 3
**Presentation:** 3
**Contribution:** 2
**Rating:** 5
**Confidence:** 4

**Summary:**

This paper investigates the many-shot in-context learning (ICL) capabilities of state-of-the-art multimodal foundation models. The authors benchmark multiple multimodal foundation models (GPT-4o, Gemini 1.5 Pro, Llama 3.2-Vision and InternLM-XComposer2.5) across 14 datasets covering multiple domains (natural, medical, remote sensing and molecular images) and tasks (image classification, visional question answering and object localization). Benchmark results indicate that the close-source models especially Gemini 1.5 Pro achieves substantial performance from many-shot ICL, while open-source models do not exhibit the same benefits.

**Strengths:**

+ The overall paper is well organized and easy to follow.
+ The idea of many-shot ICL is promising and interesting. This paper first benchmarks the many-shot ICL capabilities of state-of-the-art multimodal foundation models.
+ The dataset construction procedure is well designed, such as the diverse domain, various task and the class-balanced sampling.

**Weaknesses:**

- This paper benchmarks a limited number of models. There are only two open-source and two close-source models are evaluated in the main text. I doubt that it is hard to draw a general and convincing conclusion from such limited assessed models.
- The ablation experiments are only conducted on Gemini 1.5 Pro. Such analysis is not convincing enough, since some phenomena could be distinct when different models are utilized.
- Although this paper observes some interesting phenomena, but their deeper causes are not analyzed and explained, e.g., the reason why Llama 3.2-Vision and InternLMXComposer2.5 do not benefit from the demonstrating examples. It would be better to add more insightful analysis highlighting the future direction.

**Questions:**

Please refer the Weaknesses.

---

### Meta-Review · Area_Chair_93ta · 2024-12-18

**Metareview:**

This paper obtains four negtive scores from all reviewers. Lacking of in-depth analysis is a common concern and the authors do not resolve this issue. Most of reviewers further point out that this paper is a technical report rather than a paper. Meanwhile, the link posted in the response to reviewer DvdU also violates the double blind strategy, which should be desk rejected.

**Additional Comments On Reviewer Discussion:**

This paper obtains four negtive scores from all reviewers. Lacking of in-depth analysis is a common concern and the authors do not resolve this issue. Most of reviewers further point out that this paper is a technical report rather than a paper. Meanwhile, the link posted in the response to reviewer DvdU also violates the double blind strategy, which should be desk rejected.

---

### Decision · Program_Chairs · 2025-01-22

Reject